# Predicting the Unseen: A Novel Dataset for Hidden Intention Localization in Pre-abnormal Analysis

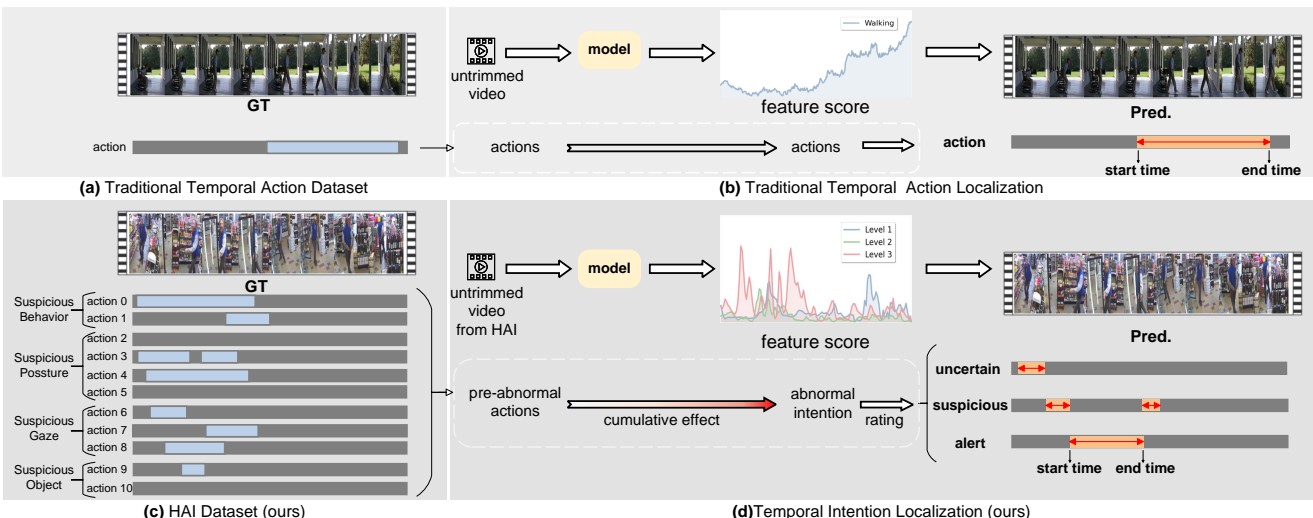

Figure 1: Comparison for task and dataset. (a) and (c) show the comparison between traditional Temporal Action datasets and our Hidden Abnormal Intention (HAI) Dataset. Traditional datasets solely concentrate on actions that are visibly occurring, but our HAI dataset analyzes the same video sample according to the structure from 4 coarse-grained dimensions as "Suspicious Gaze", "Suspicious Posture", "Suspicious Object" and "Suspicious Behavior" and 11 fine-grained perspectives to provide comprehensive guidance on abnormal intention and abnormal behavior. (b) and (d) show the comparison between traditional Temporal Action Localization (TAL) task and our Temporal Intention Localization (TIL) task. TAL predicts actions from actions. Models are provided with video input to locate the start and end times of actions by predicting action feature scores. However, our TIL predicts abnormal intention and rates intention degrees from pre-abnormal actions with analysing their cumulative effect. TIL does not directly locate isolated visible actions, instead, it focuses on locating a series of associated behavior related to abnormal intention, predicting abnormal degree (uncertain, suspicious, alert) to facilitate the judgment and prediction of abnormal behavior.

## ABSTRACT

Our paper introduces a novel video dataset specifically designed for **Temporal Intention Localization (TIL)**, aimed at identifying hidden abnormal intention in densely populated and dynamically complex environments. Traditional Temporal Action Localization (TAL) frameworks, focusing on overt actions within constrained temporal intervals, often miss the subtleties of pre-abnormal actions that unfold over extended periods. Our dataset comprises 228 videos with 5790 clips, each annotated to capture fine-grained actions within ambiguous temporal boundaries using a Joint-Linear-Assignment methodology. This comprehensive approach enables detailed analysis of the evolution of abnormal intention over time. To address the detection of subtle, hidden intention, we developed the Intention-Action Fusion module, an creative approach that integrates dynamic feature fusion across 11 behavioral subcategories, significantly enhancing the model's ability to discern nuanced intention. This enhancement has led to performance improvements of up to 139% in specific scenarios, dramatically boosting the model's

sensitivity and interpretability, which is crucial for advancing the capabilities of proactive surveillance systems. By pushing the boundaries of current technology, our dataset and methodologies foster the development of proactive surveillance systems capable of preemptively identifying potential threats from nuanced behavioral patterns, encouraging further exploration into the complexities of intention beyond observable actions.

## KEYWORDS

Multimedia, Hidden Intention, Pre-abnormal, Temporal Localization

## 1 INTRODUCTION

Temporal Action Localization (TAL) focuses on analyzing untrimmed long videos to identify and categorize periods of key action, distinguishing them from non-action backgrounds by marking the start and end times of these actions. Common datasets such as THUMOS14 [11], ActivityNet [7] and Kinetics [13] have been important in advancing this field, leading to the development of sophisticated

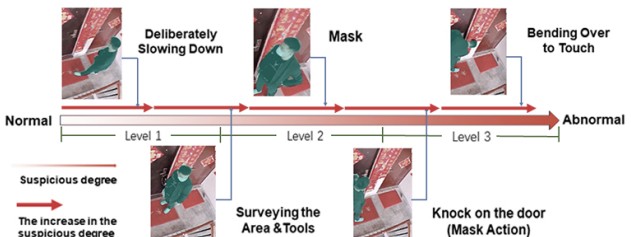

**Figure 2: As illustrated, when singular behavior, which may appear normal in isolation, accumulate, the abnormal intention gradually becomes apparent. As suspicious behavior accumulates, the suspicious degree continues to increase.**

models such as Actionformer [35], TriDet [20] [19], and Video-Mamba [3]. While these models excel in recognizing actions, they often struggle to interpret the hidden intention of these actions. This challenge becomes more apparent in situations requiring a nuanced understanding of the participants' motives.

Based on the above challenges, our research pioneers the concept of Temporal Intention Localization (TIL), aimed to identify and locate segments within untrimmed videos that potentially hidden intention, focusing on forecasting from action clues rather than merely recognizing ongoing actions.

Unlike traditional temporal action localization tasks that locate actions as they occur based on singular behavior, TIL anticipates future actions by analyzing intention combined with sequences of related behavior, thus requiring a higher degree of interpretability. The evolution of TIL underscores not only the advancements in action localization but also highlights the ongoing need to bridge the gap between recognizing actions and understanding intention. As shown in figure 1, in security-related contexts, our research tries to discover hidden abnormal intention to make intervention measures more precise and effective, facilitating pre-abnormal analysis.

The limitations of traditional coarse-grained datasets are particularly pronounced when addressing the nuanced detection of hidden abnormal intention. For instance, behavior preceding theft, like looming, may seem harmless in everyday life but can signify premeditated abnormal intention in specific contexts. Relying solely on singular, overt actions for judgment falls short in accurately detecting hidden intention. This shortfall underscores the inadequacy of existing coarse-grained labels, which lack the nuance and specificity needed to discern the subtle cues of premeditated abnormal actions. Hence, the challenge lies in understanding the context and the myriad of small, seemingly innocuous behavior that, when pieced together, reveal a larger, more concerning picture of potential abnormal behavior like figure 2.

Building upon the identified gaps in understanding the hidden intention behind actions in abnormal scenarios, our research introduces the **Hidden Abnormal Intention (HAI)** Dataset. Our dataset is designed not just to capture actions but to delve into the hidden intention that precede abnormal behavior. Unlike datasets that prioritize overt action localization, HAI maps a series of multi-perspective, multi-level behavior to their underlying abnormal intention, creating a connected narrative that highlights the causal relationship and cumulative effect between actions enhancing interpretability. Meanwhile, we value the diversity of annotator perspectives and align them with expert knowledge for a comprehensive understanding. This progression entails equipping these models with the ability to differentiate between normal behavior and those imbued with covert motives, thereby furnishing a more refined methodology for elucidating the hidden intention of humans prior to committing the abnormal.

In our paper, we undertake a detailed study to uncover hidden abnormal intention by carefully creating a dataset and refining models. Leveraging the expertise of professionals, we have constructed a fine-grained, multi-level labeling system to explore the feature of hidden abnormal behavior. Our dataset comprises 228 video, carefully annotated to reflect the gradation of abnormal intention across different scenarios, utilizing a method known as Joint-Linear-Assignment. This approach links annotated segments end-to-end before applying a linear scoring of suspicion levels to behavior, highlighting the increased suspiciousness of repetitive actions. This layered approach to labeling facilitates the development of a nuanced understanding of pre-abnormal actions.

Our detailed annotation framework directs our model improvement efforts towards locating 11 specific subcategories of behavior with abnormal intention, instead of directly identifying hidden intention segments. By aggregating predictive results for these subcategories using a weighted combination, our model finely discerns the complex web of intention preceding abnormalities. This dual approach enhances action localization precision and provides novel insights into preemptive abnormal behavior identification, significantly advancing behavioral analysis and security measures.

In our study, we advance temporal intention localization in pre-abnormal analysis with 3 key innovations:

- **Beneficial Dataset.** We introduce a specialized dataset, named HAI, aimed at understanding actions and their underlying intention related to the abnormal, enriched with expert annotations for comprehensive behavioral insights.
- **Specific Annotation.** We employ a Joint-Linear-Assignment method for nuanced abnormal intention analysis across video segments, enhancing behavioral understanding through a focus on repetitive actions.
- **Solid Method.** Our approach refines predictive models by focusing on pre-abnormal actions, improving accuracy in locating hidden intention for effective security applications.

## 2 RELATED WORK

**Temporal Video Localization Datasets.** Video datasets are crucial for advancing action recognition and temporal localization. To enhance model generalizability, many datasets include a wide range of everyday action categories, tracing back to early datasets like KTH [17] and Weizmann [5] [1]. With technological advancements, more challenging datasets have been introduced, such as UCF101 [22], Kinetics [13], ActivityNet [7], and FineAction [15]. These datasets expand action variety, enhance scene diversity, and include annotations for temporal locations and spatiotemporal bounding boxes. Rising demands for temporal localization in specific domains and unique action categories have led to the emergence of domain-specific video datasets. For example, the UCF

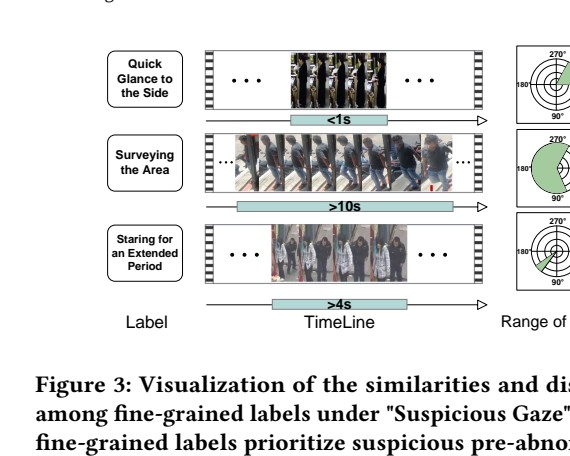

**Figure 3: Visualization of the similarities and distinctions among fine-grained labels under "Suspicious Gaze". All three fine-grained labels prioritize suspicious pre-abnormal gaze, yet diverge in range of gaze and duration.**

Sports [16] [23] and Sports-1M datasets [12], by focusing on complex sports actions, provide richer and harder research material. In more detailed action recognition (AR) research, the FineGym [18] dataset focuses on gymnastics, and the EPIC-KITCHENS [4] dataset focuses on actions and object interactions in kitchen environments, contributing to domain-specific research.

Despite efforts to improve annotation content and accuracy, most domain-specific datasets still use traditional methods for annotation. These datasets typically employ broadly related labels for annotating target actions (e.g. True/False), lacking in-depth annotations tailored to specific tasks within a given domain. In contrast, our Hidden Abnormal Intention Dataset (HAI) creatively adopts a more targeted and domain-task-aligned annotation approach: (1) HAI identifies underlying intention through sequences of causally linked actions, rather than directly annotating target behavior; (2) To capture hidden intention, we employ associative labels that cumulatively contribute to illustrating and understanding these covert objectives. (3) HAI incorporates diverse annotator perspectives to counter the subjectivity in interpreting hidden intention, enhancing accuracy through algorithmic and expert validation. (4) HAI employs a unique hierarchical labeling system that breaks down abnormal intention into 11 specific behavior analyzed from four angles, enabling precise assessment of intention intensity.

**Methods for Temporal Action Localization.** In Temporal Action Localization (TAL), deep learning has driven significant advancements. Techniques range from anchor-based methods like SSN [37] to actionness-guided approaches like UTS [28] for pinpointing actions. Additionally, graph-based models like G-TAD [34] and Transformer-based systems such as Actionformer [35] have further refined the understanding of complicated video relationships. Alongside, two-stage methods and end-to-end learning, including I3D [2] [8], offer robust frameworks for action proposal and feature extraction. Weakly supervised learning and 3D Convolutional Networks, like C3D [25], also contribute by minimizing annotation needs and capturing dynamic spatiotemporal patterns, marking a comprehensive evolution in TAL.

**Hidden Intention Detection.** In traditional intention detection, "intention" is equated with undisguised behavior, which has led to

the development of numerous intention detection algorithms. E3D-LSTM [30] predicts human intention by learning explicit spatiotemporal visual representations, while EM-base [32] and HAO [31] infer human intention by combining visual representations with attention. Automatic planning techniques, generating plans from sequences of past actions to predict possible future actions for intention identification, have been successfully applied in human intention recognition, such as the RACNN algorithm [29]. Not limited to this, HFD [33] introduces non-verbal cues as signals for intention recognition. However, these works have not explored the subtle, concealed intention in human activities, where the actor hides certain behavior—namely hidden intention. It wasn't until the introduction of the Hidden Intention Discovery (HID) task [38] that a deeper exploration into the recognition of hidden intention was highlighted. Our HAI dataset provides more fine-grained, systematic annotation materials for the recognition of hidden intention, supporting the development of more precise algorithm research, improving existing algorithms, and introducing temporal localization methods into hidden intention recognition, thus achieving a more accurate quantification and localization of hidden intention.

## 3 KEY DEFINITIONS AND CONCEPTS

Our research focuses on the complex concept of hidden abnormal intention and temporal intention localization.

### 3.1 Hidden Abnormal Intention

Hidden Abnormal Intention is defined as a series of preparatory or exploratory actions exhibited by individuals before committing abnormal actions. These actions, like subtle and indirect body language or environmental interactions, are designed to conceal true intention. For example, a potential thief may pretend to browse goods while looking for opportunities to steal. By enabling security personnel to take preventative measures beforehand, identifying these hidden intention is crucial for preventing abnormal actions.

### 3.2 Temporal Intention Localization

Temporal Intention Localization specifically targets untrimmed video to locate and identify segments indicative of abnormal intention. Given an untrimmed video, denoted as $V$, the objective is to identify and delineate a set of segments, represented as $\{s_1, s_2, ..., s_n\}$. Each identified segment $s_i$ is defined by a tuple $(t_{start}, t_{end}, l)$, where $t_{start}$ and $t_{end}$ indicate the segment's commencement and conclusion times within the video $V$, respectively. Additionally, $l$ denotes the segment's hidden abnormal intention level, categorized into three distinct levels: 1 for low, 2 for moderate, and 3 for high, each reflecting the inferred degree of hidden abnormal intention. This task aims for precise localization and accurate intention assessment, enhancing the analysis of video data for security and surveillance purposes.

## 4 HAI DATASET

Abnormal actions such as theft pose a global threat to societal security and personal property, yet tracking and investigating these acts is challenging. Effective prevention and timely identification of abnormal intention are crucial to reducing crime. This dataset is developed to assist researchers and technology developers in

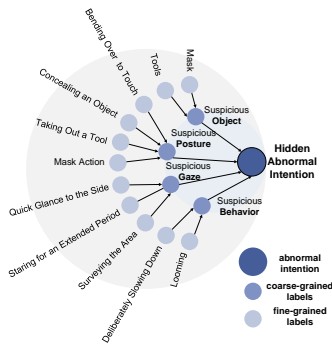

**Figure 4: Illustration of the hierarchies of HAI labels. Abnormal intention is dissected into four coarse-grained dimensions, under which eleven detailed perspectives are distinguished.**

creating systems that can recognize and locate abnormal intention, particularly during the crucial hidden phase before an incident. By leveraging data-driven insights, this initiative aims to enhance security measures and reduce risks associated with abnormal actions, thus promoting safer communities and protecting personal assets.

## 4.1 Hierarchical Labeling Framework

To capture the complexity and subtlety of HAI, we combined expert knowledge with public insight, collaborating with frontline police officers and security experts. This collaboration not only enriched our dataset with practical accuracy and value but also enhanced its broad applicability by integrating public understanding. We established a hierarchical labeling system with multi-grained labels. Coarse-grained labels identify broader behavioral categories preceding abnormal actions, such as "Suspicious Gaze," which depict general patterns during the preparatory phase. Fine-grained labels offer detailed insights, specifying actions indicative of hidden intention like "Surveying the Area" or "Staring for an Extended Period". This approach helps illuminate the complicated behavior suspects use to conceal their intention, as shown in figure 3.

The labels logically arrange the abnormal process, enhancing our understanding and detection of HAI. For example, under the coarse category of "Suspicious Gaze", we might find fine-grained labels like "Quick Glance to the Side" or "Surveying the Area" which provide insights into the specific actions a suspect might take to assess their surroundings before committing. This annotation system aims to cover the entire process from preparation to attempted actions comprehensively, revealing the complexity of abnormal intention.

In the initial of the HAI dataset, we have carefully curated and amalgamated 11 fine-grained behavior categories from a broad spectrum of hidden abnormal intention. These categories include: quick glance to the side, surveying the area, staring for an extended period, bending over to touch, concealing an object, taking out a tool, mask action, looming, deliberately slowing down, tools, and mask. These fine-grained categories are organized under four coarse-grained categories that represent different aspects of suspicious behavior: suspicious gaze, suspicious posture, suspicious behavior, and suspicious objects. This organizational framework forms the semantic backbone of our dataset, as illustrated in figure 4.

## 4.2 Dataset Collection and Annotation

Our dataset development pipeline is a structured process that spans from the initial video collection to the final integration of annotated data. The subsequent figure 5 provides a high-level overview of the process, highlighting the systematic approach taken at each stage to ensure the creation of a robust dataset for Hidden Abnormal Intention (HAI) analysis. In the following subsections, we will delve into the details of the dataset collection and the annotation steps that form the backbone of our methodology.

*4.2.1 Dataset Collection.* We sourced our videos from publicly datasets like UCF-Crime [24] and videos collected from the internet. Our dataset cover a variety of scenarios, including indoor and outdoor abnormal actions, abnormal actions involving vehicles, both individual and collaborative abnormal actions, and incidents occurring at different times throughout the day. This diverse collection is crucial because the abnormal doesn't happen in just one kind of place or time but occurring under numerous conditions.

To create an inclusive and representative dataset, we included videos with varying resolutions and frame rates, ranging from low-resolution videos common in less economically developed regions to high-definition footage from more prosperous areas. This approach ensures our dataset reflects the universal nature of abnormal actions, thereby enriching our insights.

Upon assembling the videos, we strictly excluded any footage that did not accurately capture acts of abnormal behavior or attempted incidents. This selective approach ensures the integrity and relevance of our dataset, focusing exclusively on behavior that merits further analysis.

*4.2.2 Dataset Annotation.* The entire annotation process, emphasizing multiple checks and expert feedback, ensures the dataset's high quality and utility. We believe this meticulous and strict annotation work provides researchers and practitioners with a powerful tool for better understanding abnormal behavior.

**Annotation Preparation.** To guide our annotation team, we designed a comprehensive guideline, ensuring everyone clearly understands the annotation process, the specific meanings and scenarios for each label. This guideline covers everything from video selection, importing information, to examples of each label, aiming for a standardized execution as much as possible.

**Preliminary Annotation.** In this phase, annotators apply predefined labels to the video content by identifying key behavior and accurately marking relevant time segments. Each video is thoroughly annotated by at least three individuals to ensure comprehensive label application and understanding.

**Annotation Review and Integration.** After the initial annotation, a dedicated team reviews the work to ensure accuracy and consistency. Once all annotations and reviews are complete, we compile all the information and start the integration process. We take both the union and intersection of the data annotations, preserving clips that any annotator considered suspicious (forming annotation L) and those all annotators agreed were suspicious (forming annotation S). This method allows us to simulate different levels of sensitivity to hidden abnormal intention, ensuring our dataset offers multiple angles on observing hidden intention.

**Joint-Linear-Assignment for pre-abnormal intention analysis.** In developing the annotations of hidden abnormal intention, we initiated by assigning a suspicion score to each category label per frame, adhering to a linear scale to indicate increased suspicion with repeated behavior. We then aggregated the suspicion scores

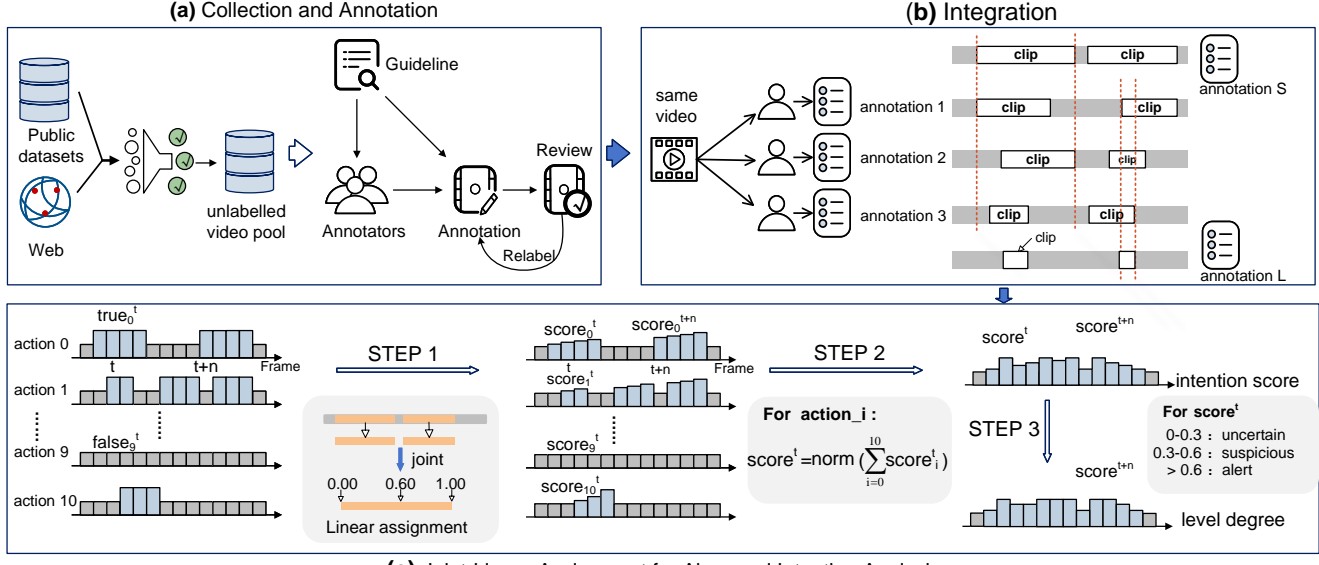

**Figure 5: Dataset pipeline overview. (a) Collection and annotation process. Our dataset compiles data from global public datasets and online abnormal actions videos, followed by expert-guided annotation by three annotators. (b) Integration process. By union and intersection on three sets of annotations, we obtain lenient (annotation L) and strict (annotation S) annotations respectively. (c) Process of Joint-Linear-Assignment for abnormal intention analysis. This process is divided into three steps: STEP 1 assigns values to the frames where an action occurs. STEP 2 calculates the intention score by summing up across 11 categories. STEP 3 converts the score into the level of abnormal intention degree.**

from all categories within a single frame, summing these values and normalizing, to accurately gauge the intensity of hidden abnormal intention for each frame. Frames with scores ranging from 0 to 0.3 are categorized as having an abnormal degree of uncertain. The true intention is not fully determined but observation should be remained. Those between 0.3 and 0.6 as suspicious, there are signs or evidence to arouse initial suspicion so increased surveillance may be conducted. Scores exceeding 0.6 are designated as alert, evidence is sufficient to indicate a high risk of abnormal so we need issue an alert or take action to prevent abnormal. This multi-level framework of hidden abnormal intention annotations provides a better understanding of the severity and potential risk associated with each observed incident.

### 4.3 Unique Challenge

In addressing the complex task of identifying hidden abnormal intention, our project faces nuanced challenges that distinguish it from standard behavior analysis tasks. These challenges highlight the complexity of our work and emphasize the need for a dataset specifically designed to capture the subtleties of abnormal behavior. Below, we delve into these unique challenges.

**Integrated Causality and Cumulative Labeling.** By annotating behavior causally related to abnormal intention and recognizing label interconnections, our dataset highlights the cumulative effect of actions, including repeated behavior and simultaneous actions. This approach reveals how behavior accumulate to manifest HAI, providing deeper analysis than traditional datasets.

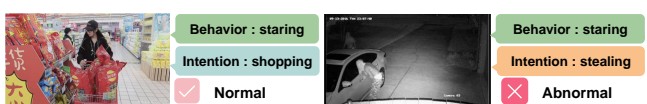

**Figure 6: For the illustrated examples, despite the apparent similarity in "staring" behavior, the underlying intention starkly differ. Our dataset aims to unearth the hidden suspicious intention beneath seemingly identical behavior.**

**Multi-Level Intention Localization and Granular Decomposition.** Our dataset offers an unprecedented level of detail, classifying hidden abnormal intention across multiple levels and dissecting these into 11 distinct behavior from four different viewpoints. This not only allows for a nuanced understanding that surpasses the simple binary categorizations found in traditional datasets but also provides an complex mapping of the gradual buildup of covert actions. Such granularity is crucial for accurately interpreting the complex interplay of micro-behavior that collectively signal abnormal intention, ensuring a comprehensive analysis of HAI.

**Inter-Annotator Variability with Expert Comparison.** We acknowledge and preserve the diversity in annotators' judgments, providing a richer, context-dependent interpretation of behavior. By comparing these diverse perspectives with expert knowledge, our dataset permits a more refined assessment of the most appropriate understanding of HAI.

**Behavior-Intention Association.** Distinguishing between simple, overt actions and hidden abnormal intention is essential. As

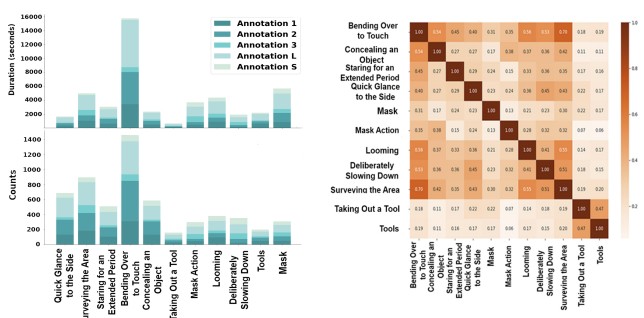

(a) duration and count of 11 labels   (b) correlation between 11 labels

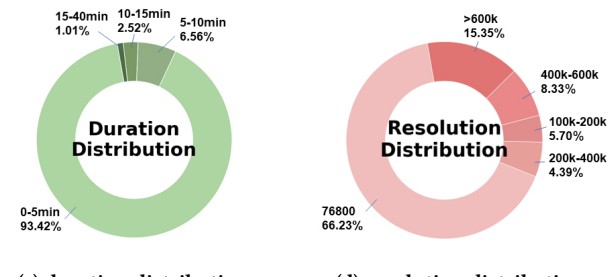

(c) duration distribution            (d) resolution distribution

**Figure 7: (a) is the duration distribution and the count distribution across five sets of annotations for eleven different labels, (b) is a heatmap displaying correlation between eleven behavior through Jaccard [9] similarity, (c) (d) are the duration and resolution distribution of videos in the HAI dataset.**

Shown in figure 6, HAI involves interpreting micro-behavior and intention, a complex task not commonly addressed in conventional behavior studies.

These challenges highlight the complexity and necessity of developing a dataset tailored to HAI, aiming to advance our understanding and detection capabilities in the area.

### 4.4 Statistics

Our HAI dataset Precisely annotates 228 videos, yielding 5 annotation files with a total of 5,790 clips spanning 8 hours of video from global datasets, the internet, and surveillance footage. It includes segments with 28 to 320 instances across 11 categories, and temporal durations from 8 to 2,223 seconds. Each video averages 122.65 seconds, includes 3.61 categories, and contains about 6 instances per clip, with instances averaging 7.4 seconds and ranging from 0.1 to 312 seconds. The dataset features diverse global locations and resolutions, authentically representing abnormal incidents. Figure 7 provides foundational data, and the heatmap calculates the Jaccard [9] similarity between label pairs in the HAI, showing more frequent label combinations in darker cells. We used the Apriori algorithm to identify common action combinations, with Table 2 listing the top three by frequency and their support values, indicating the likelihood of these combinations occurring. This analysis reveals potential patterns in hidden abnormal intention, enhancing

our understanding of behavior subtly indicating abnormal intention and focusing on high-frequency action combinations.

Our annotation framework employs a three-tier hierarchical structure with cumulative effect tagging to decode intention behind abnormal activities. In video annotations, each frame can show up to 7 behavior instances, with a cap of 17 recurrences of the same behavior in a single video. Table 1 provides statistical data, comparing our dataset with other related video datasets.

## 5 EMPIRICAL STUDIES

In the empirical study, we introduce and evaluate our novel Intention-Action Fusion (IAF) module, alongside the pioneering HAI dataset, within the realm of temporal intention localization. Through a series of rigorous experiments, including SOTA baseline comparisons and a specific ablation study, we aim to underscore the effectiveness of our methodological advancements and the intrinsic value of our dataset. This comprehensive examination not only showcases the precision and interpretability of our model in temporal intention localization but also provides a clear demonstration of how fine-grained and causality labels and dynamic feature fusion significantly contribute to the understanding of hidden intention. The empirical evidence gathered from these studies serves to highlight the crucial role of our dataset and the IAF module in advancing the field of intention analysis.

### 5.1 Intention-Action Fusion Module

We present an creative Intention-Action Fusion (IAF) Module based on underpinned by the insight that the detection of surreptitious abnormal behavior transcends the mere analysis of actions or the prediction of intention in isolation. It receives feature information from the temporal localization model on 11 types of behavior indicative of hidden abnormal intention. It extracts temporal information about key actions from the videos and then performs dynamic feature fusion on these key behavior based on predefined prior knowledge. Specifically, for each segment of video features, a weighted sum $S_{b,t} = \sum_{k=1}^{11} W_k \cdot T_{b,k,t}$ is computed for the first 11 types of behavior, where $W_k$ represents the weight for the $k$-th behavior and $T_{b,k,t}$ represents the feature value of the $k$-th behavior at time $t$ for the $b$-th video. This weighted sum is then dynamically added to the features of the intention categories of each segment across all time points, modifying the original feature tensor to $T'_{b,l,t} = T_{b,l,t} + S_{b,1,t}$ for the intention level $l$, applicable across all time points $t$. The fused information is subsequently incorporated into the feature set for hidden abnormal intention, allowing the model to flexibly optimize intention predictions tailored to the specific circumstances of different video contents. This methodology ensures that the model's predictions remain highly accurate and strongly interpretable across diverse scenarios.

### 5.2 Experiment

In this section, we initially evaluate our HAI dataset by benchmarking it against representative methods in the task of temporal action localization, demonstrating its efficacy. Additionally, we conduct an illustrative study using the Intention-Action Fusion module for temporal localization to further establish the dataset's utility. Our focus centers on the application of fine-grained labels for in-depth

**Table 1: Comparison between the HAI and other related datasets. Our dataset employs a hierarchical labeling approach. We are the first dataset to consider the cumulative effects and associative relationships of labels, adopting a hierarchical annotation format while preserving the diversity among annotators.**

| DataSet | Avg Instance Duration | Task | Hierarchical Labeling | Label Relationship | Label Interpretability | Annotator Discrepancy | Annotation Form | Year |
|---|---|---|---|---|---|---|---|---|
| HMDB51 [14] | 3.15s | AR | — | — | × | Minimize | True/False | ICCV 2011 |
| UCF101 [22] | 7.21s | AR | — | — | × | Minimize | True/False | 2012 |
| JHMDB [10] | < 2s | AR | — | — | × | Minimize | True/False | ICCV 2013 |
| THUMOS'14 [11] | 4.48s | TAL | — | — | × | Minimize | True/False | 2014 |
| ActivityNet [7] | 1.5s | TAL | 4 (Nature, Setting) | — | × | Minimize | True/False | CVPR 2015 |
| Charades [21] | 12.8s | TAL | — | — | × | Minimize | True/False | ECCV 2016 |
| Kinetics400 [13] | 10s | TAL | — | — | × | Minimize | True/False | 2017 |
| AVA [6] | 2.7s | TAL | — | — | × | Minimize | True/False | CVPR 2018 |
| FineGym [18] | < 2s | TAL | 2 (Time), 3 (Semantic) | — | × | Minimize | True/False | CVPR 2020 |
| HACS [36] | 33.2s | TAL | — | — | × | Minimize | True/False | CVPR 2021 |
| EPIC-KITCHENS [4] | 3.1s | TAL | — | — | × | Minimize | True/False | IJCV 2022 |
| Our Dataset | 7.4s | TIL | 3 (The Behavioral Vehicle of Intention) | Cumulative | ✓ | Retain | Level | 2024 |

**Table 2: Top three combinations by occurrence frequency at different levels. We can study the potential behavioral patterns in abnormal behavior through this information.**

| Level | Combination | Support |
|---|---|---|
| uncertain | Surveying the Area, Bending Over to Touch | 0.0463 |
| | Bending Over to Touch, Concealing an Object | 0.0421 |
| | Quick Glance to the Side, Surveying the Area | 0.0386 |
| suspicious | Surveying the Area, Bending Over to Touch | 0.0544 |
| | Mask, Bending Over to Touch | 0.0490 |
| | Quick Glance to the Side, Surveying the Area | 0.0418 |
| alert | Surveying the Area, Bending Over to Touch | 0.1326 |
| | Quick Glance to the Side, Surveying the Area | 0.1254 |
| | Mask, Bending Over to Touch | 0.1147 |

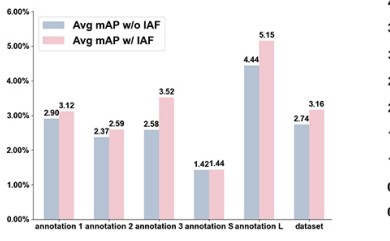
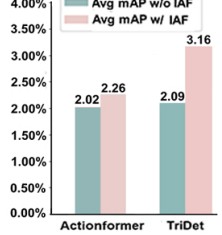

(a) Ablation study of IAF module     (b) Overall performance

**Figure 8: (a) Ablation study of our IAF module for different annotations on TriDet [20] [19]. (b) Performance of overall dataset on Actionformer [35] and TriDet [20] [19].**

learning and understanding, specifically investigating how such detailed annotations facilitate the identification of covert behavior with the aid of the Intention-Action Fusion module. The aim is to highlight the crucial role our HAI dataset and methodological enhancements play in accurately detecting clandestine actions.

*5.2.1 Implementation Details.* In our study, we aim to detect and locate hidden abnormal intention, thus enabling the proactive identification of abnormal incidents within video. To achieve this, we conduct a detailed evaluation of leading temporal recognition approaches at various levels of granularity, leveraging two SOTA : TriDet [20] [19], ActionFormer [35]. We conducted experiments on a dataset with five distinct sets of annotations. These sets include annotation 1, annotation 2, annotation 3, annotation S, annotation L. Annotation 1 was contributed by the least experienced annotator, annotation 2 by the most experienced, and annotation 3 by an annotator with moderate experience. Additionally, we utilized the annotation integration method to obtain strict annotation (annotation S) and lenient annotations (annotation L). The performance of the models on these annotations gives us insight into how experience in annotation may influence the outcome of machine learning models. The dataset is divided into training and testing sets with a 9:1 ratio. We utilize VideoMaeV2 [26] [27] to extract video features.

*5.2.2 Evaluation Metrics.* In our analysis, we measure the accuracy of located action intervals using Intersection over Union (IoU), a key metric that quantifies the overlap between the predicted action

intervals and the ground truth. IoU is determined by dividing the size of the overlap between the ground truth interval (G) and the predicted interval (D) by the size of their union, expressed as $|G \cap D| / |U|$. The performance evaluation considers a range of IoU thresholds from 0.3 to 0.7, allowing for a comprehensive assessment of prediction accuracy across different levels of strictness.

*5.2.3 Results.* Table 3 and figure 8a presents the complex challenges embedded within the HAI dataset, revealed through the lens of temporal intention localization tasks. The diverse annotations spectrum necessitates a nuanced understanding of the dataset's inherent complexities. Prior to the integration of the Intention-Action Fusion (IAF) module, the Actionformer [35] model displayed a range of Avg mAP percentages from 0.59% to 3.39%, indicating that the difficulty of intention localization for different experiential annotated data in the dataset is variable. Therefore, in order to reduce individual subjective bias and comprehensively evaluate the performance of the model on the dataset, we simultaneously calculate the overall dataset Avg mAP by taking the average of the model's results on five annotations. For the Actionformer [35] model with the IAF module, Avg mAP values range from 0.79% to 3.37% across different annotations, with an improvement in the dataset Avg mAP from 2.02% to 2.26%. Incorporating the IAF module brings about an overall performance boost, especially in annotation

**Table 3: Comparative performance metrics of the Actionformer [35] and TriDet [20] [19] models on the HAI dataset with and without the Intention-Action Fusion (IAF) module. The table showcases mean Average Precision (mAP) scores at varying intersection over union (IoU) thresholds (0.3 to 0.7) for different annotation categories.**

| Backbone | annotation | IAF module | mAP@0.3 | mAP@0.4 | mAP@0.5 | mAP@0.6 | mAP@0.7 | Avg mAP |
|---|---|---|---|---|---|---|---|---|
| Actionformer [35] | annotation 1 | × | 5.57 | 3.85 | **2.37** | 0.90 | **0.33** | 2.60 |
| | | ✓ | **5.76** | **4.52** | 2.28 | **1.24** | 0.28 | **2.82**↑ |
| | annotation 2 | × | **5.07** | **2.62** | 1.32 | **0.80** | **0.36** | **2.03** |
| | | ✓ | 3.02 | 2.14 | **1.35** | 0.71 | 0.30 | 1.50 |
| | annotation 3 | × | 3.49 | 2.18 | 1.09 | 0.55 | 0.09 | 1.48 |
| | | ✓ | **5.60** | **4.11** | **2.87** | **0.91** | **0.65** | **2.83** ↑ |
| | annotation S | × | 1.42 | 0.79 | **0.49** | **0.19** | **0.07** | 0.59 |
| | | ✓ | **2.25** | **1.02** | 0.45 | 0.14 | **0.07** | **0.79**↑ |
| | annotation L | × | 7.04 | **5.59** | 2.55 | 1.15 | 0.65 | **3.39** |
| | | ✓ | **7.11** | 4.21 | **3.02** | **1.55** | **0.96** | 3.37 |
| TriDet [20] [19] | annotation 1 | × | 5.25 | 3.82 | 2.53 | 1.31 | 0.62 | 2.71 |
| | | ✓ | **5.30** | **3.86** | **3.16** | **2.28** | **1.00** | **3.12**↑ |
| | annotation 2 | × | 4.45 | 2.63 | 1.55 | 0.77 | 0.21 | 1.92 |
| | | ✓ | **5.00** | **3.69** | **2.63** | **1.09** | **0.53** | **2.59**↑ |
| | annotation 3 | × | 3.30 | 2.28 | 1.10 | 0.49 | 0.18 | 1.47 |
| | | ✓ | **6.66** | **5.06** | **3.90** | **1.51** | **0.50** | **3.52** ↑ |
| | annotation S | × | 1.94 | 1.43 | 0.65 | 0.12 | 0.03 | 0.83 |
| | | ✓ | **2.57** | **2.00** | **1.33** | **1.09** | **0.21** | **1.44** ↑ |
| | annotation L | × | 7.26 | 4.88 | 2.92 | 1.97 | 0.59 | 3.52 |
| | | ✓ | **9.82** | **7.12** | **4.96** | **2.45** | **1.42** | **5.15** ↑ |

3 with a more moderate level of experience that aligns with the general public, Avg mAP has improved by 91.21%.

In concert with the findings from the Actionformer [35] model, the TriDet [20] [19] with the IAF module showcases a span of Avg mAP values ranging from 1.44% to an impressive 5.15%, culminating in a dataset Avg mAP of 3.16%. The incorporation of the IAF module has led to a substantial enhancement in model performance, with Avg mAP improvements oscillating between 15% and a remarkable 139% across the five annotations. Mirroring the trends observed with the Actionformer [35], the deployment of the IAF module within the TriDet [20] [19] model notably excelled in annotation 3, underlining the module's adeptness in optimizing localization in scenarios with a balanced complexity that mirrors the collective cognitive baseline of the general populace.

The use of the IAF module in various annotations has demonstrated how deep learning models can effectively reflect common human experiences through data annotation. Particularly in annotations that combine expert knowledge with everyday experiences, the IAF module effectively refines the model, improving its ability to recognize complex intention. This discovery highlights the value of creating data annotations that connect with a wide range of human experiences. By enhancing the model's performance on annotations that incorporate professional expertise and everyday details, the IAF module makes significant progress in narrowing the divide between artificial intelligence and the complex world of human actions and intention.

*5.2.4  Ablation experiment.* To further elucidate the vital contribution of our proposed Intention-Action Fusion (IAF) module to the model's performance, we designed a specific ablation study. This process involved using the TriDet [20] [19]model as our baseline and integrating the IAF module with an essential modification: during the model's actual learning phase, the features used and optimized did not pass through the IAF module. However, the model

still maintained its focus on learning pre-abnormal actions, effectively simulating the scenario where the IAF module's advanced feature processing capabilities were bypassed. As vividly illustrated in figure 8b, the outcomes of this study were telling. There was a noticeable decrement in performance post-ablation, a phenomenon that was particularly pronounced in the dataset's Avg mAP metric. This stark contrast underscores the indispensable role of the Intention-Action Fusion module in amplifying the model's acumen for locating hidden intention. It is a step forward in our endeavor to refine and push the boundaries of hidden intention localization methodologies, aiming to achieve unprecedented levels of accuracy and efficiency in understanding and interpreting complex, nuanced intention embedded within actions.

## 6  CONCLUSION

In this paper, we propose a new task—temporal intention localization, which involves the chronological positioning of individuals' intention, and introduce an novel dataset named the Hidden Abnormal Intention Dataset (HAI). Unlike other existing datasets, the HAI creatively employs a marking scheme more tailored to temporal intention localization, utilizing associative labels with a cumulative effect, diversity in annotation, and hierarchical labeling. This provides a solid foundation for algorithmic research in temporal intention localization. Building upon HAI, we have also developed a unique algorithm to address the challenges of temporal intention localization, and the results highlight the significant challenges posed by our dataset for this task. These challenges underscore the difficulty and importance of recognizing hidden abnormal intention in natural settings. We hope our efforts will pave the way for new advancements in the field of temporal intention localization, thereby contributing to the assurance of public safety and societal well-being.

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
