# OpenReview forum: "Predicting the Unseen: A Novel Dataset for Hidden Intention Localization in Pre-abnormal Analysis"
_acmmm.org/ACMMM/2024/Conference — MM2024 Poster_

### Official Review · Reviewer_SVyP · 2024-05-17

**Rating:** 3
**Confidence:** 3

**Summary:**

This paper introduces a dataset for the Temporal Intention Localization (TIL) task, designed to identify hidden abnormal intentions in videos of complex environments. Compared to the traditional Temporal Action Localization (TAL) task, TIL can proactively identify potential threats. Additionally, this paper proposes the Intention-Action Fusion module, which can be integrated into existing TAL methods to enhance TIL performance. Notably, the author suggests that if the dataset could be open-sourced, it would have significant implications for the further development of TIL tasks.

**Strengths:**

1. This paper is easy to follow, the motivation and formulation of the paper are sound
2. The HAI dataset, annotated using Joint-Linear-Assignment, provides multi-level classification of intentions, aiding in the proactive identification of potential threats
3. Extensive ablation experiments have demonstrated the effectiveness of the IAF module

**Limitations:**

1. In the Dataset Collection section, the authors describe, "We sourced our videos from publicly datasets like UCF-Crime [24] and videos collected from the internet." However, further description of the data from the internet is needed, along with consideration of user privacy in the videos
2. The lack of detailed description regarding data annotation, such as the mention of generating 5 annotation files, including 2 from post-processing, raises questions. Does this imply that annotation was conducted by three groups of annotators with different expertise distributions? Within each group, were multiple annotations made for the same video? If only one person annotated, does it introduce the possibility of individual bias
3. Reviewing annotated data is equally important for assessing the quality of the dataset. The paper lacks information on the proportion of relevant reviews and the pass rates of annotated data through review. Additionally, are multiple rounds of annotation and review conducted during the annotation process? If multiple rounds of annotation exist, when is the annotation considered complete
4. Although the effectiveness of the IAF module has been demonstrated on ActionFormer and TriDet, its motivation is unclear

**Suitability:**

2

---

### Official Review · Reviewer_ypqZ · 2024-05-23

**Rating:** 4
**Confidence:** 2

**Summary:**

In this paper, the authors collect a new video dataset annotated with fine-grained temporal labels regarding intention prediction. They proposed a model based on a behavior category fusion module to detect subtle intentions.

**Strengths:**

The authors provide a clear description of the motivation behind the Hidden Abnormal Intention （HAI） problem and have conducted extensive temporal and class annotations.

The focus of this dataset differs significantly from existing mainstream datasets on action detection and video content understanding, which serves as a potential basis for many research topic.

**Limitations:**

Although the authors annotate the data with fine-grained temporal information, multi-level categories, and continuous abnormal intention values, the annotations are not sufficient in terms of the number of videos reflecting the overall scene distribution, the number of clips reflecting local intention trends, and the duration of instances. This can lead to models learning biases in the distribution (such as imbalances in fine-grained category labels). Particularly regarding the duration of instances, in the intention prediction task proposed by the authors, richer contextual information has a more comprehensive indicative capability for intentions.

**Suitability:**

2

---

### Official Review · Reviewer_afTW · 2024-05-25

**Rating:** 4
**Confidence:** 2

**Summary:**

The paper introduces the Hidden Abnormal Intention (HAI) dataset, designed to identify hidden abnormal intentions in complex environments. The dataset emphasizes pre-abnormal actions and their cumulative effects, leading to abnormal behavior. The authors propose the Temporal Intention Localization task, which focuses on predicting abnormal intentions rather than visible actions. The paper also presents the Intention-Action Fusion module, enhancing the model's ability to detect nuanced intentions through dynamic feature fusion across multiple behavioral subcategories.

**Strengths:**

1. The paper introduces the Hidden Abnormal Intention (HAI) dataset, designed to identify hidden abnormal intentions in complex environments. The dataset is meticulously annotated, capturing fine-grained actions and cumulative effects, offering a detailed understanding of abnormal behavior.
2. The HAI dataset fills a significant gap by focusing on hidden abnormal intentions, providing a rich resource for future research.
3. Experimental validation and benchmarking against state-of-the-art models showcase significant performance improvements.

**Limitations:**

1. Some paragraphs are poorly structured, with single-sentence paragraphs that disrupt the flow of the manuscript, such as lines 616.
2. Table 1 should explicitly compare the dataset size (e.g., number of videos) with similar datasets to provide a fuller comparison.
3. More references to related research on intention prediction, such as pedestrian intention, should be included to provide context and background for the work.
[2023] Pit: Progressive interaction transformer for pedestrian crossing intention prediction
[2023] Faster pedestrian crossing intention prediction based on efficient fusion of diverse intention influencing factors
4. The paper should clarify if the dataset will be publicly accessible upon acceptance, and provide more details on how it will be shared (e.g., via Google Drive or a project homepage) and any access restrictions.
5. The paper should address potential ethical risks, particularly regarding the inclusion of abnormal behaviors and personal biometric information in the videos, and provide more details on how these issues are handled.
6. The authors should include more examples of the dataset in the supplementary materials to give readers a fuller understanding of the data.
If the authors address my concerns, I would be happy to rejust my score.

**Suitability:**

2

---

### Official Review · Reviewer_Z1pi · 2024-05-26

**Rating:** 4
**Confidence:** 2

**Summary:**

This paper introduces a novel video dataset designed for Temporal Intention Localization (TIL) aimed at identifying hidden abnormal intentions in densely populated and dynamically complex environments. Traditional Temporal Action Localization (TAL) frameworks focus on overt actions within constrained temporal intervals and often miss the subtleties of pre-abnormal actions unfolding over extended periods. The dataset comprises 228 videos with 5790 clips, each annotated using a Joint-Linear-Assignment methodology to capture fine-grained actions within ambiguous temporal boundaries. By integrating dynamic feature fusion across 11 behavioral subcategories, the proposed method significantly enhances the model's ability to discern nuanced intentions, thereby advancing proactive surveillance systems.

**Strengths:**

Innovative Dataset Design: The paper introduces the Hidden Abnormal Intention (HAI) dataset, which not only captures actions but also delves into the underlying abnormal intentions. The dataset includes rich fine-grained annotations and multi-dimensional behavior labels, providing a comprehensive framework for behavior analysis.

New Temporal Intention Localization Task: The introduction of the TIL task aims to predict abnormal intentions by analyzing sequences of related behaviors rather than just recognizing ongoing actions. This extends the research scope beyond traditional TAL tasks and improves proactive analysis capabilities in security systems.

Feature Fusion Method: The development of the Intention-Action Fusion (IAF) module, which dynamically integrates features from 11 behavioral subcategories, significantly enhances the sensitivity and interpretability of the model in identifying hidden intentions.

**Limitations:**

In section 5.1, even if this paper describes a dataset, the corresponding model figure should be shown, especially for models proposed for a new dataset.

For the experimental results in Table 3, even though the model proposed in this article effectively enhances the performance of each method on the new dataset, such a low average performance raises doubts about its practical application significance.

**Suitability:**

2

---

### Meta-Review · Area_Chair_VWKr · 2024-07-05

**Recommendation:** Accept (Poster)
**Confidence:** 4

**Metareview:**

The paper introduces a new task called Temporal Intention Localization, along with a corresponding dataset named Hidden Abnormal Intention (HAI) that includes detailed action categories. Additionally, it presents an Intention-Action Fusion module designed to improve the detection of subtle intentions by dynamically fusing features across multiple behavior subcategories. The paper received positive scores from 3 out of 4 reviewers. While some negative comments focused on the annotation accuracy due to unclear annotation procedures, the overall contribution of the proposed task and dataset surpass the acceptance threshold.

Clarity: The paper is well-written and is easy to read and follow.

Originality: The novelty of this work is sufficient for acceptance.

Significance： The proposed task can motivate research of Interpretable abnormal detection, and has the potential to inspire other video understanding tasks. The constructed dataset is a contrbituion to the community.